# Behavior of Cows in the Lying Area When the Exit Gates in the Pens Are Opened: How Many Cows Are Forced to Get Up to Go to the Milking Parlor?

**DOI:** 10.3390/ani13182882

**Published:** 2023-09-11

**Authors:** Marek Gaworski

**Affiliations:** Department of Production Engineering, Institute of Mechanical Engineering, Warsaw University of Life Sciences, 02-787 Warsaw, Poland; marek_gaworski@sggw.edu.pl; Tel.: +48-22-593-45-83

**Keywords:** behavior, cow, free-stall housing system, lying, milking, pen

## Abstract

**Simple Summary:**

In barns that use milking parlors, cows walk from their pens to the milking area at least twice a day. A question can be raised: In what position are the cows at the moment of opening the exit gate in the pen, before going to the milking parlor? Is it a standing or lying position? When the cow is lying down, she must stop this activity. The forced break from lying can be considered in terms of cow welfare. The study compared the number of cows lying down when the exit gates opened in four pens over a period of 26 days. A greater number of lying cows were found before leaving for morning milking compared to afternoon milking. More cows were lying on the stalls with a lower level of sand compared to a higher level of sand. For comparisons between animals, an index of forced standing up of cows has been proposed. The results of the research may be an inspiration to identify solutions that will reduce the forceful getting up of cows. Such solutions include milking cows with a voluntary milking system instead of a milking parlor.

**Abstract:**

Equipping a farm with a milking parlor requires moving groups of cows from their pens to the part of the barn where milking takes place. The task of moving cows, carried out two or three times each day, shows links to the lying area of the barn. When the cows are taken from the pen to the milking parlor, some of them may be lying down, and this lying down must be interrupted. The forced standing up of cows can be considered in terms of their welfare. The aim of the study was to examine the number of cows lying in the stalls at the time of opening the exit gates in the pens in order to take the cows to the milking parlor. The study covered four pens, each with 12 cows. The behavior of the cows in the pens before morning and afternoon milking was recorded over 26 days. In the analysis, the dependent variable was the number of lying cows, and the independent variables were the time of milking and the level of sand in the lying stalls. The results of the study showed the significance of differences in the number of lying cows for stalls with a low and high level of sand, both in the case of morning and afternoon milking. Differences in the number of lying cows were also found when comparing the time before morning and afternoon milking. To compare the tendency of individual cows to lie down before going to milking, an index of forced standing up was proposed. The research conducted showed differences in the behavior of cows before leaving the pen to the milking parlor. The stage to reduce the forced standing up of cows is to equip the farm with an automatic milking system (AMS) instead of using a milking parlor. In barns with AMS, cows have full freedom to get up and approach the milking stall. The results of the observations are thus an additional argument confirming the benefits of using an automatic milking system, considered in terms of the welfare of dairy cows, regarding their lying down and getting up.

## 1. Introduction

In the technology of dairy cattle production in livestock facilities, key importance is attached to keeping cows in a free-stall housing system. The importance of the free-stall housing system is confirmed by its comparison with the tie-stall housing system. One of the main areas of comparison is animal welfare [1]. The free-stall housing system, compared to the tie-stall system, has significant potential to achieve a high level of animal welfare in the barn. This is due to greater freedom of movement and unrestricted expression of behavior typical of cattle, as well as joint integration within a group of animals. Welfare reflects how well animals cope with their environment [2]. Assimilation, achieved with little resources and effort, is an indicator of a satisfactory level of animal welfare [3]. 

For most parameters within the four welfare principles, Popescu et al. [4], based on the Welfare Quality Protocol, identified significant differences between free-stall and tie-stall dairy housing. Their research has shown that the free-stall system is better for cattle in terms of feeding, housing conditions, and the expression of natural behavior. The ability to express the natural behavior of cows in barns with a free-stall housing system translates into easier identification of cows in estrus and insemination at the right time [5]. Housing in a free-stall system creates a positive stimulus for cows to express their natural behavior in the lying area. These behaviors concern the possibility of choosing the most comfortable resting conditions, taking into account the quality of the bedding [6] and the structural features of the lying stalls, including the dimensions of the stalls [7,8], positioning of the neck rail [9,10], and the equipment of the stalls with brisket boards [11]. Examples of looking for the comfort of rest in a barn with a free-stall housing system are also the choice of cows in a row with lying stalls in a pen [12] and lying places in a row of stalls [13]. A key factor identifying the link between the lying area and the welfare of dairy cattle is lying time [14].

In practice, the lying time of cows is understood and considered in studies, e.g., [15,16,17] as the number of hours per day spent by cows resting in the lying area (space). The assessment of the time spent lying by cows applies to both free-stall and tie-stall barns. However, there may be more detail to the interpretation of lying time than just the number of hours spent lying down in a 24 h period. This is confirmed by the undertaken research, which includes a comparison of the lying time of cows during the day and at night, additionally in connection with different seasons [18,19] and climatic conditions [20,21], a comparison of the lying time of healthy and sick cows, including those with mastitis [22] and lameness [23], or lying time of cows during heat stress [24,25]. 

Many factors determine the lying time, and the behavior of cows in the lying area is part of the assessment of cow welfare. Achieving welfare is conducive to the full freedom of expression by dairy cattle of their needs and behaviors, including those related to lying down and getting up. However, in barns with a free-stall housing system equipped with a milking parlor, the full freedom of cows to lie down may be disturbed in practice. This disturbance of the freedom of lying is due to the need to leave the pen to go to the milking parlor. Thus, the assessment of lying cows is the result of technology, which provides for the need to leave the pen two or three times and go to the milking parlor. 

If the cows are in a standing position before going to the milking parlor, they are ready to leave the pen. Lying cows have to get up, so do one extra action to get out of the pen. Their lying activity is interrupted due to the need to go to the milking parlor. Hence, the following research problems can be formulated: (1)What proportion of cows in the group are forced to get up before leaving the pen to go to the milking parlor?(2)Does the time of milking (morning or afternoon) affect the number of cows that need to be forced to stand up in the stall?

The aim of the research study was to assess the behavior of dairy cows in the lying area in the period immediately before the animals leave the pen for the milking parlor. 

The study of the behavior of cows in the pen is part of the broadly understood issue of assessing the welfare of ruminants. The issue of welfare assessment is an area of interdisciplinary research that combines a biological factor (animals) with technical, technological, and herd management factors. In the process of cattle welfare mapping [26], it remains important to look for places and situations related to welfare assessment. Such a situation occurs when cows leave the pen for the milking parlor. The assessment of such a situation may be the subject of research considerations that fill the gap in the current state of knowledge. 

## 2. Materials and Methods

The research was carried out at the Dairy Education and Research Centre in Agassiz (BC, Canada), i.e., the experimental farm of the University of British Columbia in Vancouver. During the research period (November/December), the average air temperature outside the barn ranged from 1 to 5 °C. The animals were managed in accordance with the standards (guidelines) of the Canadian Council of Animal Care [27]. 

### 2.1. The Area of Observation and Its General Characteristics 

Observations of cow behavior were carried out in a barn with a free-stall housing system. It was a naturally ventilated barn with side walls fitted with roll-up curtains. The barn consisted of 24 modular pens. There were 12 lying stalls in each of them. Thus, there were a total of 288 lying stalls in the barn, where mainly dairy cows and partly (in four pens) heifers over 1.5 years of age were kept. During the study period, some of the pens in the barn were connected in such a way that they formed modules with 24 stalls and 36 stalls. Four modular pens, i.e., each with 12 lying stalls, were selected for detailed observations. It was a series of adjacent pens located in one part of the barn. Each pen had access to a feeding alley that ran along the pens. There were two feeding alleys in the barn. The element separating the pen from the feeding alley was a neck-rail feeding ladder. In each pen covered by the study, the construction of the feeding ladder was the same. Feed (TMR) was distributed in the feeding alley while the cows were in the milking parlor for both morning and afternoon milking. 

Each pen had an individual gate connecting to a passageway leading along the center of the barn to the milking parlor. Through this gate, cows left the pen for the parlor and returned after milking. The width of the gate in each pen was such that it allowed free movement of the animals, both when leaving and entering the pen. The direction of opening the gates allowed the cows to enter and exit the pen undisturbed without making additional turns of the body or circling the open gate. The construction of each gate was based on a set of metal pipes. The walls separating individual pens included in the study were also constructed of metal pipes. Thus, the cows in separate, modular pens had visual contact with each other. 

The basic equipment of each pen was a lying area, comprising 12 lying stalls. The lying stalls were arranged in three rows, with four stalls in each row. The two rows were connected in such a way that the cows lay facing each other with their heads towards the common part connecting the rows of stalls. The third row was adjacent to the part of the pen bordering through a concrete wall with the passageway to the milking parlor, located in the central part of the barn. The lying stalls in a row were separated from each other by Artex Y2K partitions (Artex Fabricators Inc., Langley, BC, Canada); the full design of each stall was complemented by a neck rail. In each row, one periphery stall was adjacent to a fence separating it from an adjacent pen, while the other periphery stall had a concrete wall on one side. The total (designed) length of the stalls in double (head-to-head) rows was 240 cm. In the third row (single row), the stalls were 30 cm longer. The width of the lying stalls was, on average, 115 cm. In the front part of each stall, there was a brisket board, which limited the effective length (measured from the brisket board to the curb) to 165 cm. 

The bedding material at the lying stalls was sand. It was river sand, cleaned of organic material found in the river. The sand met quality standards for particle size; before delivery to the farm, the sand was sorted to eliminate gravel and stones that could reduce the comfort of the animals in contact with the bedding material in the stalls. After use in the barn, sand mixed with feces was removed outside the barn, and there it was segregated into two fractions: Organic and mechanical. The organic fraction was used to fertilize arable land and grassland on the farm. 

In each pen, there were two manure alleys, in which manure scrapers operated automatically (activated automatically at fixed intervals, 6 times a day). One manure alley was located between the wall separating the feeding alley and the row with lying stalls. The second manure alley covered the space between the other two rows of stalls. Due to the requirements for the quality (accuracy) of the scraper operation, the manure alleys were lowered in relation to the areas intended for the passage of animals to the drinking bowl and the exit gate. Thus, the movement of cows in the pen was associated with overcoming steps between the manure alleys and the surfaces for passing animals. 

The pens were also equipped with bath-type drinkers. The drinkers were built into the partition separating adjacent pens in such a way that one drinker was used by cows from two pens. The bath-type drinkers were equipped with an automatic water filling system, which fulfilled its tasks as a result of the water intake by the animals. 

The floor, both in the manure alleys and in the transitional parts of the pen, including the area with access to the drinker and the exit gate, was made of concrete. The concrete surface was additionally grooved in the area with access to the drinker and in front of the exit gate. 

In the space of each pen, there was a fan above the double row of lying stalls. The operation of the fan during high temperatures was supposed to reduce the risk of heat stress in cows. 

### 2.2. Scope of Observation in the Pen and Ensuring the Assumed Levels of Sand in the Lying Stalls 

The observations covered a total of 48 Holstein Friesian dairy cows. The cows were housed in modular pens with 12 animals in each; for the purposes of the analysis, the pens were marked as 1, 2, 3, and 4. The number of animals corresponded to the number of lying stalls in each pen. The cows included in the study belonged to the same technological group. Thus, they received the same TMR feed. Cows had ad libitum access to feed. Fresh fodder was supplied to the feeding wagon twice a day, at 06:00 and 16:00. Several times a day, the feed was pushed into the feeding alley with a tool carrier with an angled shovel.

Cows in the pens covered by the study were characterized by lactation (mean ± SD) 3.06 ± 2.25. During the study period, the cows showed no signs of disease or lameness. 

Observations of the cows were carried out over the following 26 days. In each group, the cows had been together for at least three days before the start of the observation. During the research period, the same cows were kept in individual pens; there was no need to replace them due to health, randomness, or other reasons. 

The observations of cows concerned their behavior in the lying area at the moment of opening the exit gate to the corridor leading to the milking parlor. The moment of opening the gates in individual pens before the departure of the cows for both morning and afternoon milking was taken into account. To accurately determine when the gates were open and how the cows behaved at that time, the area of the pens was recorded with cameras. One camera was installed above each pen included in the study, at a height of 8 m above the floor surface. Each camera was positioned to record the exit gate and lying area for the dairy cows. The cameras were automatically set to record every day between 05:00–07:00 and 15:00–17:00. The relatively wide time interval covered by the recordings resulted from the fact that, in practice, cows are not taken to the milking parlor at exactly the same time on particular days. In addition, the exit of the cows from each of the four pens to the milking parlor was shifted so that cows from separate groups would not join together. After leaving the pen, a group of 12 cows was led to a side-by-side milking parlor with two rows of milking stalls. There were 12 milking stalls in each row, which is the number of cows in one group. This solution made it easier to manage the movement of groups of animals between the pens and the milking parlor. An important role in this management was played by the waiting area (in front of the milking parlor) and the arrangement of corridors for moving animals.

A set of video cameras (WV-BP330, Panasonic, Osaka, Japan) cooperating with a digital video recording system (Genetec Inc., Saint-Laurent, QC, Canada) was used to record the research material in the pen. Based on the recorded material, the time of opening the exit gates and the behavior of cows in the lying area were identified, and this information was transferred to a form in Excel. The identified behavior of cows in the lying area was their following positions: Lying down, standing with two legs, and standing with four legs on the lying stalls. During the research period, no cases of cows lying outside the lying area were observed. A detailed analysis covered only cows lying in stalls at the time of opening the exit gates. These cows were forced to get up to go with the other cows in the group to the milking parlor. The person managing the herd in the barn was responsible for making the cows stand up. 

For the purposes of the experiment, the pens included two options for the amount of sand in the lying stalls. Pens 1 and 2 had a low sand level (10 cm below the level of the rear curb—Figure 1a), and pens 3 and 4 had a high sand level (sand level equal to the surface of the rear curb—Figure 1b). The appropriate level of sand in the lying stalls in individual pens was adjusted daily, during the afternoon milking, when the cows were in the milking parlor. The level of sand was leveled with rakes. The sand to be replenished in the stalls was stored in the part in front of the stalls to which the cows had no access. The same person was responsible for correcting and maintaining the assumed sand level during the 26 days of the experiment. 

In general, the conducted observations did not disturb the daily rhythm of the animals, including leaving the pen to the milking parlor. Both before and after the observations, dairy cows went to the milking parlor every day, which was part of herd management on a dairy farm. 

In order to analyze the selected group of research data, the index of forced standing up of cows (*IFS*) was proposed. The following formula was used to calculate this index:(1)$$IFS=\frac{N_{lc}}{N_{od}\cdot N_{m}}$$
where *N_lc_* is the total number of cases of a given cow lying down at the moment of opening the exit gate in the pen during the study period, *N_od_* is the number of observation days, and *N_m_* is the number of milkings per day. 

The definition of the index of forced standing up of cows (*IFS*) shows that its value ranges from 0 to 1. A value of 0 means that during the observation period, a given cow has not once been lying in the pen when the exit gate was opened before the cows left for the milking parlor. On the other hand, the maximum value of 1 would mean that every day and before each milking, a given cow was lying in a lying stall. The *IFS* index is calculated for individually considered cows.

### 2.3. Statistical Analysis 

The experimental unit was a cow; therefore, only results related to cow observations were included in the statistical analysis. The analysis was carried out using Statistica v.13 software (StatSoft Polska, Cracow, Poland). Descriptive statistics of the study results were carried out. Statistical analysis involved the use of a non-parametric test to compare the significance of differences in the results of observations covering the number of lying cows for given independent variables (sand level in lying stalls and milking time). The assumed significance level was α = 0.05. 

## 3. Results

The results of observations regarding the behavior of cows in the lying area at the time of opening the exit gates in the pens are presented in Table 1, taking into account the criteria of individual pens. The results were compiled on the basis of observations carried out throughout the study period, i.e., 26 days, including morning and afternoon milking. Table 1 includes the median, minimum, and maximum number of cows for each behavior and pen. The number of cows in each pen was 12; therefore, the data in Table 1 refer to a group of 12 animals. 

During the observation period covering the time of opening the exit gates in the pens, no cow was lying outside the lying area. 

The summary of observation results, which is presented in Table 1, indicates the dominant role of the identified cases of lying cows in comparison with other forms of activity in the lying area (standing with two and four legs in lying stalls). Only in the case of lying was a median of at least 1 or more found for each pen. Therefore, in the further analysis of the research results, the focus was on the assessment of cases of cows lying down at the time of opening the exit gates in the pens. The collected (recorded) research material made it possible to determine the number of cows lying in the lying stalls, taking into account the grouping variables. These variables were the time (morning or afternoon) of departure of the animals to the milking parlor and the criterion of the level of sand in the lying area. The results of the observations covering these criteria are presented in Table 2, along with the average number, standard deviation, minimum and maximum number of cows, and the number of observations. 

In the undertaken statistical analysis, the aim was to verify the hypothesis about the significance of the differentiation of the dependent variable, i.e., the number of lying cows for a given grouping variable. Both for the variable grouping the milking time and the level of sand, there was no normal distribution (examined with the Shapiro–Wilk test) of the dependent variable, i.e., the number of lying cows at the time of opening the exit gates in the pens. The condition of normal distribution was not met, so the parametric statistical test for the analysis of variance was not used.

Statistical analysis of the observation results was performed using the non-parametric Mann–Whitney test. Taking into account the assumptions of the Mann–Whitney test, which is a test for two independent samples, statistics were calculated for the grouping variable, i.e., the level of sand at the lying stalls. The dependent variable in the experiment was the number of lying cows (at the time of opening the exit gate in the pen), and so for morning milking, the statistic value was Z = 5.398 and *p* = 0.000. On the other hand, for afternoon milking, the corresponding statistical values were Z = 2.113 and *p* = 0.0346. In both cases, the *p*-value is lower than the significance level (*p* = 0.05); therefore, the difference between the number of lying cows for the two considered levels of sand is statistically significant, both in the case of morning and afternoon milking. 

The results of the analysis of data on lying cows are also presented in Figure 2, taking into account the comparison of morning and afternoon milking for the grouping variable, i.e., low and high levels of sand in lying stalls. 

The number of cows lying in stalls at the time of opening the exit gates in the pens was also assessed, taking into account the subsequent 26 days of the research study. For the purposes of this part of the analysis, four considered cases were distinguished, taking into account the behavior (lying) of cows in:–pens with a low level of sand during the morning milking,–pens with a high level of sand during the morning milking,–pens with a low level of sand during the afternoon milking,–pens with a high level of sand during the afternoon milking.

The results of the study, i.e., the number of lying cows at the time of opening the exit gates in the pens for the four considered cases, are presented in Figure 3. 

The individual values (number of lying cows) in Figure 3 are average values, calculated on the basis of data from two pens with the same level of sand. There were two pens with a low level of sand and two pens with a high level of sand in the experiment. 

In the analysis of the data presented in Figure 3, the days when there are no cows lying in the stalls at the time of opening the exit gates in the pens are crucial. If there are no lying cows in the stalls when the exit gates are opened, there is no need to interrupt the lying time of the cows. This can be interpreted in relation to the welfare of resting animals. The analysis of individual graphs in Figure 3 shows that only in one case there is no day with the number of lying cows equal to 0. This is the result of observations for the morning milking in the case of a group of cows kept in stalls with a low level of sand. In fact, for the case under consideration (morning milking—low level of sand), one day was found in one pen where no cow was lying in the lying area when the exit gate was opened. The data in the graphs is based on calculating the average number of lying cows in two pens with the same level of sand. Therefore, the calculated average value in a given case may not reflect the real situation in the pens. This situation confirms the importance of in-depth analysis of research data. The results of such in-depth data analysis are presented in Table 3.

The differences in the number of observed cases without cows lying down for the two options of approach to the analysis of the results are particularly large for the afternoon milking of cows (Table 3). A comparison of average results (for two pens) and individual pen results identified differences between pens (with the same level of sand) and the groups of cows within them. This can be confirmed by comparing the cows in the pens covered by the study.

On the basis of the collected research material, the index of forced standing up of cows (*IFS*) was calculated. The calculation of the index—according to Formula (1)—was performed for each cow, and the results were presented taking into account the criteria of individual pens (Table 4). 

The highest values of the calculated index of forced standing up of cows (*IFS*) were found for pens with a low level of sand. This applies to both mean, minimum, and maximum values (Table 4). Such calculation results reflect the previously presented comparisons of the number of lying cows in individual pens and different lying conditions (low and high levels of sand in lying stalls).

The minimum and maximum values of the *IFS* index correspond to the behavior of individual cows in the given pens. The differences between the minimum and maximum *IFS* index ranged from three times (pen 1) to more than fifteen times (pen 3). Such data indicate significant differences between cows that required forced standing before leaving the pen for the milking parlor. 

The large variation between cows in the number of cases of lying down when the exit gates in the pens were opened was an inspiration to compare the observation results with some data characterizing the cows. It was decided to check the relationship between the number of considered cases of lying cows with their DIM (days in milk) and the number of lactations. DIM was taken into account at the beginning of the study. The conducted analysis did not confirm the correlation between the number of cases of individual cows lying down at the time of opening the exit gates in pens with their DIM and the number of lactations. The coefficient of determination (R^2^) was at its maximum level below 0.3. 

## 4. Discussion

The problems of lying cows receive a lot of attention in research and scientific discussions. The benefits of lying cows justify the development of detailed research aimed at a thorough understanding of the factors determining lying down and its assessment. The aim of the presented research was to join the discussion on lying cows by showing the approach to the assessment of lying down in connection with the key task in the technology of dairy production, i.e., milking.

Research to date has been dominated by the observation of certain lying behaviors that translate into an assessment of animal welfare. Total lying time per day, bout frequency, and single bout time are among the key parameters and, at the same time, indicators for assessing the welfare of cows in the lying area [14].

The lying time of dairy cows per day is an evaluated parameter in many studies and may cover the range of 9.5–12.9 h/d per animal [15], 8.2–13.2 h/d [17], and even up to 15 h/d [16]. Observations and evaluation in the lying area are also subject to the frequency of bouts (number of bouts per day) and the average duration of one bout. In sample studies [28,29,30], the average bout frequency for most cows ranged from 9 to 11 bouts per day, and the average bout duration ranged from 60 up to 99 min. Studies have highlighted possible large variations between cows in bout frequency, ranging from less than 5 bouts/d to more than 20 bouts/d [30]. Large differences resulting from the individual characteristics of cows are also observed in the case of a single bout, which can be both very short (a few minutes) and very long—even several hours [31].

In the presented study, also related to the lying of cows, the time of lying (in hours) and the number of bouts per day, as well as the duration of a single bout, were not included in the observations. The subject of observation was the number of cows lying in pens at the time of opening the exit gates. Thus, the presented results of this research extend the current approach to the scope of research and analyses conducted in the lying area for dairy cattle. In part, the presented research refers to the bout, specifically the end of one of the bouts, when the lying cow has to get up and go to the milking parlor. In the case under consideration, it is a forced standing of the cow, i.e., breaking one of the bouts due to the need to leave the pen. The forced interruption of one of the bouts can undoubtedly be considered in the context of animal welfare. Disturbances in the free behavior (lying) of cows in the pen reduce the comfort of their rest, which certainly requires further research. At this point, it would also be worth referring to previous studies that included bouts in the evaluation of cows lying down. A question can be formulated: Do bout-observed studies count only bouts that are natural behaviors of animals or also include man-made getting up of cows before going to the milking parlor? The answer to this question may be useful in correcting the approach to assessing cow bouts. 

The lying time of cows and the number of bouts per day are generally assessed in studies against a number of factors. These are animal health and motivation, environmental, technical, technological, and herd management factors [14]. A characteristic feature of these studies is the continuous observation of animals, including the recording of cows or other methods of collecting data on their activity 24 h a day. This approach to data collection made it possible to determine the impact of production problems on the daily lying time of cows. These problems are lameness [32] and other diseases [33], including mastitis [34], the amount of bedding material [35] and its quality [6,36], the dimensions of lying stalls [7], and oversizing the cow herd [37]. In the presented study, instead of many hours of observation per day, an approach was used to record a very short period of time, including the time of opening the exit gates in the pen. This resulted from the specificity of the formulated objective and scope of the study on the link between lying down and milking. Both lying and milking are included in the time budget of a dairy herd [38,39]; hence, the study of the links between these activities determines the improvement of the assessment of dairy production technology [40]. 

The problem of the relationship between lying down and milking has already been addressed in previous studies by Österman and Redbo [41], but in this case, it was crucial to examine the effect of milking frequency (two and three times a day) on lying down and getting up of cows. In this study, the total duration of lying and standing cows as well as the lying down intention in the 4 h period before milking were compared. In the study by Charlton et al. [42], the laying of cows and their milking were additionally linked to the density of herds on farms with a free-stall housing system. However, Norring and Valros [43] added research on the motivation of cows to lie down in the context of milking in parlors, which requires waiting for milking. The time spent by cows in the waiting room before entering the milking parlor is the result of managing groups of dairy cows in the barn [44]. The subject of the relationship between lying cows and milking has been developed in the presented research, showing that milking cows has an impact on the comfort of lying down and the possibility of its disruption. 

The conducted research and its scope became the premise for proposing the index of forced standing up of cows (*IFS*). The intention of the proposed index is to identify the number of cases involving the forced standing of cows. Forced standing up of cows (to go to the milking parlor and in other situations) can be considered in terms of animal welfare assessment and welfare disruption. The *IFS* index complements the previously developed and used research indices related to the assessment of various activities of cows in the barn. The indices, which include CLI (cow lying index), CSI (cow standing index), and CFI (cow feeding index), are the basis for monitoring [45] and assessing the behavior and comfort of cows in connection with the season, time of day, and housing system [19], with particular emphasis on the free-stall housing system [46]. A key element in determining these and other indicators is the behavior of cows in different areas of the barn, identified in terms of the comfort achieved [47]. The *IFS* index also takes into account the aspect of cow comfort, but from the point of view of disturbing this comfort.

In studies of cow behavior, including the lying area, there is a reference to different daily periods [48,49]. Different periods of the day are taken into account in studies on cow behavior in parallel with selected animal characteristics, for example, body temperature [50]. Two periods of the day were taken into account in the research, but their choice resulted from the time of milking on the farm. 

The intention of many studies involving the assessment of cow behavior is to recognize animal preferences, including those related to lying, when alternative housing solutions are implemented [51]. The presented research also included an approach involving two compared options differing in the level of sand at the lying stalls. When there was more sand in the stalls, there was a smaller number of cows lying at the time of opening the exit gates in the pen. Paradoxically, a larger amount of sand in the lying area should provide higher resting comfort for cows [35] and should translate into a greater number of resting cows. Meanwhile, the situation was the opposite: In the stalls with more sand, a smaller group of cows rested compared to the stalls with less sand. The activity of the cows, in this case, could be due to the difference in distance between the neck rail and the bed with different levels of sand. The number of resting cows can be considered one of the indicators of animal welfare; a greater number of resting cows can be associated with a higher level of welfare expressed by the animals. However, in the context of the research conducted on the exit of cows to the milking parlor, it was found that the greater amount of sand in the stalls resulted in a smaller number of cows whose rest had to be interrupted due to the exit to the milking parlor. Therefore, if a smaller group of cows was interrupted from lying down, that is, a smaller number of animals had disturbed resting comfort and welfare. 

Small groups of cows were studied in this study. Each of them consisted of 12 animals. For comparison, the research should be conducted on larger groups of cows that are taken to the milking parlor. Proportionately more cows could lie in such groups during the opening of the exit gates. However, it is difficult to clearly predict what the percentage of cows lying down would be in relation to the total number of animals in the herd at the moment of opening the exit gate in the pen to go to the milking parlor. In this case, it seems crucial to pay attention to the number of cows in the herd in relation to the number of lying stalls in the pen. In large herds, it sometimes happens that the number of cows is greater than the number of lying stalls. Overstocking reduces the lying time of cows and increases their competition for lying stalls [37]. In this case, it can be expected that more cows will lie down when the exit gates to the milking parlor are opened. However, such a statement needs to be checked on the basis of observations carried out on a large herd of cows. 

In practice, apart from the observation of various forms of cow behavior in the lying area, there is also an approach based on calculating indicators characterizing the occupancy of lying stalls in a free-stall system [52]. The presented results of this research, focusing on the assessment of the behavior of cows at the moment of opening the exit gates in the pens, can be extended by the results of observation of the place (stalls) where the cows are lying in the pen. This range of research will be developed in future experiments.

Many studies on the assessment of cow behavior in the lying area, but also in other zones of the barn, emphasize the importance of the method of approach to collecting research results and their development. The issue of selecting the sampling frequency that would best characterize the daily behavior of cows in free-stall housing systems is raised [53]. The methodical approach to research also includes new methods of measuring cow activity levels in free-stall barns [54,55,56]. In the current study, a well-known method of cow recording was used, which allowed the identification of the number of animals lying down when the exit gates in the pens were opened. In practice, other solutions could also be used to collect research data, including data loggers (DL). Identification of cows lying in a pen at a given moment may be the subject of the development of research methods that will allow for the automatic collection of observation results and automatic data transfer for further analysis. 

The presented research problem concerning leaving the pen and moving cows to the milking parlor is part of the discussion on animal welfare. This discussion highlights the differences between milking cows in a milking parlor and a milking robot. These differences are also related to the evaluation of cows lying down. With a voluntary milking system, cows are free to lie down and get up when they feel the need to go to be milked. Such a situation—considered in terms of activity in the lying area—can be considered as fulfilling the principles of cow welfare. In barns where milking takes place in the milking parlor, it is sometimes necessary—as confirmed by the current study—for some cows to stand up forcibly when the whole group is going to be milked. The forced standing of cows can be considered a disturbance of animal welfare. A comparison of automatic milking systems (AMS) with milking cows in milking parlors, apart from technological and technical efficiency [57], economic [58], production performance [59], and functional aspects [60] or concerning animal health [61] and milk quality [62], should also take into account selected aspects of laying cows [63,64]. In this way, it becomes possible to complete the overall picture of the comparison between automatic and conventional milking systems. 

## 5. Conclusions

The intention of the study was to indicate the problem of cows being forced to get up from lying positions due to the need to go to the milking parlor. The interruption of lying cows in lying stalls may be considered in terms of lowering animal welfare. On the other hand, there is a need for the cows to go to the milking parlor to be milked. Hence, there is a problem of finding a compromise between the welfare of cows in lying stalls and the fulfillment of the basic technological task in dairy production, which is daily, two- or three-time milking. With such a range of considerations, it seems justified to emphasize the principles of milking cows using automatic milking systems. The voluntary milking system allows the cows to freely choose the milking time, which does not interfere with their freedom to lie down or get up. This is an additional argument confirming the benefits of implementing automatic milking systems in barns. These benefits relate, in this case, to the welfare of the lying cows in the lying area. 

## Figures and Tables

**Figure 1 animals-13-02882-f001:**
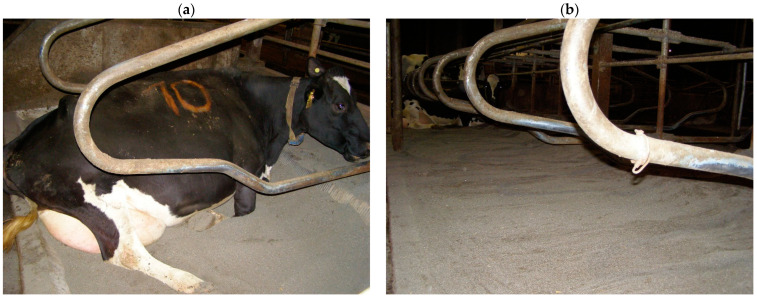
Lying stalls with a sand level 10 cm below the level of the rear curb (**a**) and sand level equal to the surface of the rear curb (**b**).

**Figure 2 animals-13-02882-f002:**
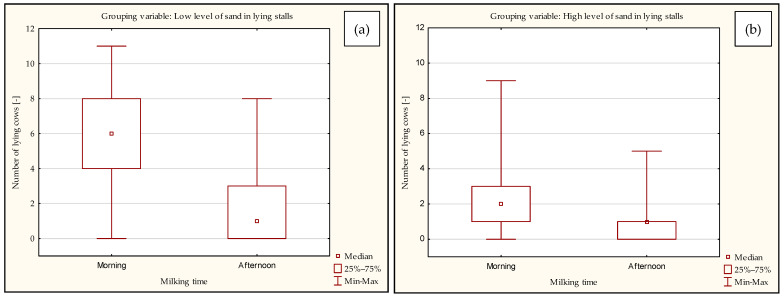
Comparison of the number of lying cows at the opening of the exit gates in the pens for the morning and afternoon milking times, taking into account the pens with a low level of sand (**a**) and high level of sand (**b**) in the lying stalls.

**Figure 3 animals-13-02882-f003:**
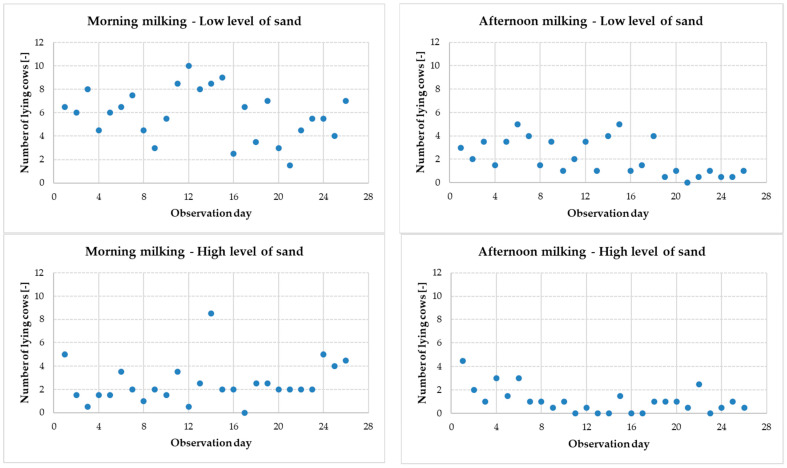
The number of lying cows at the time of opening the exit gates in the pens on the following days of observation, taking into account the time of milking and the level of sand in the lying stalls.

**Table 1 animals-13-02882-t001:** Behavior of the cows (lying, standing with two legs and four legs on the lying stalls) in the four pens, when the exit gate to the alley leading to the milking parlor is opened; number of cows in each pen: 12.

Cow Behavior	Pen Number	Median	Min	Max
Lying down on the lying stall	1	5	0	11
	2	2	0	9
	3	2	0	9
	4	1	0	9
Standing with two legs on the lying stall	1	1	0	7
	2	0	0	4
	3	0	0	3
	4	0	0	2
Standing with four legs on the lying stall	1	0	0	2
	2	0	0	0
	3	0	0	0
	4	0	0	1

**Table 2 animals-13-02882-t002:** The number of lying cows at the time of opening the exit gates in the pens for the considered criteria, i.e., the time of milking and the level of sand in the lying stalls; number of cows in each pen: 12.

Milking	Sand Level in the Lying Stalls	Number of Observations	Number of Lying Cows			
			Mean	SD	Min	Max
Morning	Low	52	5.87	3.04	0	11
	Full of sand	52	2.52	2.20	0	9
Afternoon	Low	52	2.13	2.27	0	8
	Full of sand	52	1.10	1.22	0	5
All observations		208	2.90	2.88	0	11

**Table 3 animals-13-02882-t003:** Comparison of the number of cases with no cows lying down at the time of opening the exit gates in the pens, taking into account the average of two pens (with the same level of sand) and individually considered pens.

Milking	Sand Level	Number of Cases without Lying Cows	
		Average of Two Pens (Same Level of Sand)	Individually Considered Pens
Morning	Low	0	1
	High	1	7
Afternoon	Low	1	16
	High	6	19

**Table 4 animals-13-02882-t004:** The values of the index of forced standing up of cows in the pens covered by the study.

Pen Number	Sand Level	Index of Forced Standing up of Cows—*IFS*		
		Mean	Min	Max
1	Low	0.44	0.21	0.60
2	Low	0.23	0.12	0.42
3	High	0.17	0.02	0.31
4	High	0.13	0.04	0.40

## Data Availability

Not applicable.

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
