# Peer review of "Behavior of Cows in the Lying Area When the Exit Gates in the Pens Are Opened: How Many Cows Are Forced to Get Up to Go to the Milking Parlor?"

_animals, 2023, doi:10.3390/ani13182882_

Round 1

Reviewer 1 Report

Dear author, congratulation for the manuscript. I have concern about the research design, though.

The studies of animal behaviour must be checked for non-independence such as animals influencing each other. In other words, checking the data sets for pseudo-replication because non independence may make the whole set invalid. Did the author have this checked out?

No comments. 

Author Response

Dear Reviewer,

Thank you for reviewing the article and suggestions regarding the research project. I reviewed the recorded material with cows in pens again, in terms of independence and the potential influence of animals on each other. The cows lay down on separate (individual) lying stalls, so during lying down (which is the main subject of the research) they had no physical contact with each other and did not interact with each other. Analyzing the recorded material again, I did not find any cases where the cows would interfere with each other's lying in their lying stalls. I have not identified cases where one cow would disturb another cow and, for example, cause the cow to stand up with her behavior. The cows were marked, so I could recognize them, and I could tell from the recorded material that the individual animals (the cows) didn't affect each other. I wonder how animals (cows) in a given group would influence each other. Such a situation would take place, for example, if the number of cows in the pen was greater than the number of places (stalls) to lie down. Then cows with a higher position in the hierarchy in the herd could "force" other individuals to get up when there are no free (unoccupied) lying stalls. In the study, the number of cows in each pen was the same as the number of lying stalls.

I also asked a professor from the University of British Columbia's Animal Welfare Program for additional help in assessing independence in the study groups of animals. The study presented in the article was carried out on an experimental dairy farm of this University. The professor stated: "While there is a certain number of animals in the pen, the animals are free to lie down and some have to be forced to get up before going out of the pen into the milking parlor".

Reviewer 2 Report

This paper might be shortened and should be highly clarified. The assessment that cows are "forced" to stand up should be clairified as well. A detail : the author might choose between "parlour" and "parlor". This is a minor point in regard of the rewriting that might possibly be undertaken.

Many sentences are very abstract and wordy. Very difficulet to read and follow the arguments.

Author Response

Dear Reviewer,

Thank you for reviewing the article and all suggestions. I corrected the wording in the article; only the words "parlor" and "behavior" are used, which I have marked in red in the text. I clarified (explained) in the text the problem of forced getting up of cows before going to the milking parlor (lines: 210-213).

I admit that the descriptions, especially in the methodological part, are very detailed, which gives the impression that they are boring and difficult to follow. My intention was to present the conditions of the experiment as precisely as possible. This is what I learned during my research and multiple stays at the UBC Dairy Education and Research Center in Agassiz.

An alternative was to include in the article a diagram presenting the layout of the four pens, including lying stalls, corridors and other details. However, I have come to the conclusion that more details can be provided through the description.

I made corrections in the text. I removed some sentences, especially in the part concerning the research methodology. I tried to break long sentences into shorter ones without losing the meaning of the information presented. In general, I prefer writing short sentences and clearly presenting information, because it is easier to reach the reader this way. I use this principle in both scientific and popular science articles.

Reviewer 3 Report

It is a very interesting work related to the behaviour of cows in the lying area and the influence of forcing the cows to get up to go to the milking parlour. The article is well organized and well written and, as such, is simple to follow. Accurate references support the text. The Tables are essential for understanding the article. The material and methods are clearly described, which allows a perfect understanding of what has been done. The results are well presented and well discussed. Finally, the results corroborate the conclusions.

Some detailed comments are below:

Throughout the text, the spelling is a non-British variant. Please change the title a check the manuscript for consistency. Behaviour of cows in the lying area when the exit gates in the pens are opened: How many cows are forced to get up to go to the milking parlour? "change with" Behavior of cows in the lying area when the exit gates in the pens are opened: How many cows are forced to get up to go to the milking parlor? Along

L47 has a significant potential in achieving a high "change with" has significant potential to achieve a high

L77 Many factors determining the lying "change with" Many factors determine the lying

The English is fine

Author Response

Dear Reviewer,

Thank you for reviewing the article. I have introduced the suggested language changes in the text. I standardized the spelling of the words "parlor" and "behavior", taking into account the title of the article and its content.

Thank you for your kind words regarding the article and the research idea that I implemented in the study.